

# Soil respiration across a variety of tree-covered urban green spaces in Helsinki, Finland

Esko Karvinen[1], Leif Backman[1], Leena Järvi[2,3], and Liisa Kulmala[1,4]

[1]Climate System Research, Finnish Meteorological Institute, Helsinki, Finland
[2]Institute for Atmospheric and Earth System Research (INAR), University of Helsinki, Finland
[3]Helsinki Institute of Sustainability Science (HELSUS), Faculty of Science, University of Helsinki, Finland
[4]Institute for Atmospheric and Earth System Research (INAR), Forest Sciences, University of Helsinki, Finland

**Correspondence:** Esko Karvinen (esko.karvinen@fmi.fi)

**Abstract.** As an increasing share of the human population is being clustered in cities, urban areas have swiftly become the epicentres of anthropogenic carbon (C) emissions. Understanding different parts of the biogenic C cycle in urban ecosystems is needed in order to assess the potential of enhancing their C stocks as a cost-efficient means to balance the C emissions and mitigate climate change. Here, we conducted a field measurement campaign over three consecutive growing seasons to

examine soil respiration carbon dioxide ($CO_2$) fluxes and soil organic carbon (SOC) stocks at four measurement sites in Helsinki representing different types of tree-covered urban green space commonly found in northern European cities. We expected to find variation in the main drivers of soil respiration – soil temperature, soil moisture, and SOC – as a result of the heterogeneity of urban landscape, and that this variation would be reflected in the measured soil respiration rates. In the end, we could see fairly constant statistically significant differences between the sites in terms of soil temperature but only

sporadic and seemingly momentary differences in soil moisture and soil respiration. There were also statistically significant differences in SOC stocks: the highest SOC stock was found in inactively managed deciduous urban forest and the lowest under managed streetside lawn with common linden trees. We studied the impacts of the urban heat island (UHI) effect and irrigation on heterotrophic soil respiration with process-based model simulations, and found that the variation created by the UHI is relatively minor compared to the increase associated with active irrigation, especially during dry summers. We conclude

that, within our study area, the observed variation in soil temperature alone was not enough to cause variation in soil respiration rates between the studied green space types, perhaps because the soil moisture conditions were uniform. Thus, irrigation could potentially be a key factor in altering the soil respiration dynamics in urban green space both within the urban area and in comparison to non-urban ecosystems.

## 1   Introduction

Urbanisation and climate change are two topical themes in current discussion on the human-nature relationship. Over 55 % of the global population lived in urban areas in 2018 and that percentage is likely to increase in the near future (Das, 2021). Urban areas are notable sources of atmospheric carbon dioxide ($CO_2$) (Pataki et al., 2006; Canadell et al., 2009; Velasco and Roth, 2010), and since the most recent trend of rapid increase in atmospheric $CO_2$ concentration is due to human activity





(Arias et al., 2021), many cities are currently setting up climate programs with the aim of carbon (C) neutrality in the coming

years or decades (European Commission, 2022). Carbon neutrality can be achieved by reducing C emissions, compensating for them, or maintaining and increasing C sinks and stocks in urban vegetation and soil, the last of which is often deemed the most cost-efficient option (Faivre et al., 2017).

When considering the different C stocks in nature, soil organic carbon (SOC) stock is of especial interest because of its vast quantity: estimates of global SOC stock range between 1500–3000 Pg C (Eswaran et al., 1993; Scharlemann et al., 2014) - a

magnitude which clearly exceeds the estimated global organic C stocks in aboveground vegetation or in the atmosphere (Lal, 2004; Scharlemann et al., 2014). SOC stock is formed by C inputs from aboveground and belowground litter, root exudates, and possible organic amendments (Davidson and Janssens, 2006; Basile-Doelsch et al., 2020). Even though only 2.7 % of global terrestrial soils are urban (Lal and Stewart, 2018), by utilising judicious management practices urban ecosystems have potential to sequester and store C in soil and vegetation on a local scale (Lal and Augustin, 2012; Foldal et al., 2022), which

benefits the aforementioned C neutrality goals of cities and municipalities.

However, the current understanding of biogenic C cycle in urban environments is mostly based on dynamics observed in more intensively studied non-urban ecosystems such as forests and agricultural lands. Urban ecosystems differ from non-urban ecosystems in terms of light availability, temperature, precipitation and water cycle, pollution, restrictions in soil volume and crown space, and the level of human-induced disturbance (Sæbø et al., 2003; Kaye et al., 2006), all of which have an impact on

urban biogenic C cycle (Lal and Augustin, 2012). The urban heat island (UHI) effect, caused by anthropogenic heat sources and heat stored and re-radiated by built structures, elevates air temperature in urban areas compared to their non-urban surroundings (Oke, 1982; Rizwan et al., 2008). The UHI effect also creates temperature variation within the urban area because of varying building density and the heterogeneity of land cover and land use types that comprise the urban landscape (Yan et al., 2014; Edmondson et al., 2016; Lan and Zhan, 2017; Johnson et al., 2020). Some urban green spaces are irrigated for various reasons

during the growing season (Ignatieva et al., 2020; Cheung et al., 2021; Pan et al., 2023) which makes their soil moisture conditions notably different from areas under natural precipitation.

Many urban green spaces are constructed, during which their soil and other growing media are established based on multiple parallel needs. The land use history of a specific urban green space can be diverse and the lifespan of its current state not necessarily so long. As a result, there often is no evident coupling between the aboveground vegetation and the belowground

C in urban green spaces that is often found in more naturally developed ecosystems (e.g. Frouz et al., 2009; Pinno and Wilson, 2011; Dantas et al., 2020); observed SOC stock tends to represent the decisions made and actions taken while establishing the particular green space rather than reflect the current aboveground vegetation and its dynamics.

Soil respiration ($R_S$) is the $CO_2$ flux from soil surface to atmosphere as a result of belowground plant and microbial respiration (Ryan and Law, 2005), and it is the second largest terrestrial carbon flux (Bond-Lamberty and Thomson, 2010; Lei

et al., 2021). It can be further classified into autotrophic ($R_A$) and heterotrophic ($R_H$) respiration in which the former originates from plants and their roots, and the latter from fungi, bacteria, and animals living in soil and litter (Burba, 2022). In practice, $R_S$ is the key pathway through which C transfers from SOC stock to the atmospheric C stock as SOC is decomposed by microbial activity (Davidson and Janssens, 2006). Soil temperature and moisture are important controls for $R_S$ (Bond-Lamberty



and Thomson, 2010; Burba, 2022), and the SOC stock size itself also affects the decomposition rate (Davidson and Janssens,
60    2006).

Measurement-based estimates of SOC stocks in urban green space have been reported in previous literature and shown to
vary across climatic conditions. In cold and temperate climates, the estimates for SOC stock in urban parks range between
9.7–35.5 kg C m$^{-2}$ depending on the aboveground vegetation type, management type, and park age (Pouyat et al., 2006;
Dorendorf, 2014; Setälä et al., 2016; Lindén et al., 2020; Cambou et al., 2021). Areas with the most intensive management
practices have been reported to have the highest SOC stocks and these may be more than two times larger (per area) than in
natural grasslands and agricultural lands (Pataki et al., 2006; Golubiewski, 2006). Two studies conducted in Helsinki (Finland)
also observed high SOC stocks (19.5 kg C m$^{-2}$) in park soils under the most intensive management class (Setälä et al., 2016;
Lindén et al., 2020).

Previous studies measuring urban $R_S$ are more scarce than estimates of SOC stock, but some indicators for specifically
urban dynamics exist. Decina et al. (2016) measured $R_S$ in urban soils in Boston (USA) finding up to 2.2 times higher $R_S$ than
measured in the closest rural ecosystems. However, Weissert et al. (2016) observed that urban $R_S$ in Auckland (New Zealand)
was similar to non-urban forests and grasslands. Incorporating compost in urban soils, that is increasing their SOC stock, was
shown to increase $R_S$ in Liverpool (UK) (Beesley, 2014). Goncharova et al. (2018) reported that in their measurements in
Moscow (Russia) soil temperature was an important control for $R_S$ in spring and autumn, whereas soil moisture was the main
controlling factor during summer, when soil temperature was above 10 °C, which could imply that irrigation plays a significant
role in summer. Wu et al. (2016) demonstrated how in Beijing (China) $R_S$ was elevated at the boundary between urban green
space and impervious surface as a result of higher soil temperature. Conversely, $R_S$ at urban forest edges in Boston has been
shown to be reduced due to higher temperature and more probable aridity (Garvey et al., 2022); a phenomenon contrasting
what has been observed in non-urban forests in Petersham (USA) (Smith et al., 2019).

The above, seemingly contradictory, examples demonstrate the need to i) further characterise urban SOC stocks and $R_S$
dynamics, ii) consider urban ecosystems separately from non-urban ecosystems, and iii) take into account the variation in
environmental conditions within the urban area. In this study, we aimed to answer those needs by analysing $R_S$ and its drivers
in urban green space, focusing on the following research questions:

1. Can we distinguish differences in soil respiration rates measured in different types of tree-covered urban green space? If
yes, are the differences explained only by consistent differences in soil moisture, soil temperature, or SOC stocks?

2. To what degree does the UHI affect heterotrophic soil respiration rate during the growing season?

3. To what degree does irrigation affect heterotrophic soil respiration rate during the growing season?

To answer these questions, we carried out a field measurement campaign in four different types of tree-covered urban
green spaces in Helsinki, over three consecutive growing seasons. Additionally, we used process-based ecosystem modelling
to specifically answer research questions 2 and 3. We hypothesised that we would find different levels of soil moisture, soil
temperature, and SOC across the green space types included in this study due to the heterogeneous urban environment, and



that these differences would also be reflected in differences in $R_S$ rates. We also hypothesised that the UHI effect alone would have a notable effect on the $R_S$ rate in urban ecosystems and that irrigation would allow the $R_S$ rate to remain at a higher level throughout the growing season than would be the case in non-irrigated environments under natural precipitation.

## 2 Material and methods

### 2.1 Site description

This study was conducted in Helsinki, the capital of Finland, which in 2020 had a population of 656 920 (1 524 489 for the whole metropolitan area) and a population density of 3020 people per km$^2$ of land area (City of Helsinki, 2021). Average annual temperature and precipitation were 6.5 °C and 653 mm, respectively, during the reference period of 1991–2020 (Finnish Meteorological Institute, 2022). Almost 34 % of the city's land area in 2021 (the total of which was 217 km$^2$ including inland waters) consisted of green space managed by the city (City of Helsinki, 2021). Our four measurement sites were located in the Kumpula and Hermanni districts in central Helsinki (Figure 1). They encompassed a variety of green space types commonly found in northern European cities: an urban forest (Forest), a fruit garden (Orchard), a managed park (Park), and a road verge between a roadway and a sidewalk (Streetside).

The Forest site was situated at the edge of a small urban forest patch with silver birch (*Betula pendula* Roth) the dominant tree species. Other deciduous trees such as downy birch (*Betula pubescens* Ehrh.), Norway maple (*Acer platanoides* L.), and Scots Elm (*Ulmus glabra* Huds.) formed the subcanopy. Understory vegetation was sparse and consisted mainly of ground elder (*Aegopodium podagraria* L.). The Orchard site was comprised of apple trees (*Malus domestica* Borkh.) growing on a managed lawn. The lawn was mown manually a few times each summer and was not irrigated or fertilised. The Park site was located within the Kumpula Botanic Garden and consisted of four small-leaved linden trees (*Tilia cordata* Mill.) growing on a managed lawn. The lawn was mown daily by a mowing robot, was fertilised once every few years, and was irrigated during dry periods. However, the mowing robot could not access the section of the lawn on which the measurements were conducted; the lawn there was mown manually a few times each summer. The Streetside site was a row of common linden trees (*Tilia x europaea* L.) growing on a strip of managed lawn between a roadway and a sidewalk. The lawn was mown manually a few times each summer and was not irrigated or fertilised. Mean tree trunk diameter (diameter at breast height, DBH) at all sites was in a similar range (20–30 cm) but the standing tree volume was largest at the Forest site because of the distinctively taller trees (22 m) compared to the other sites (6.5–12.5 m) (Table A1). Some further descriptions of the sites can be found in Ahongshangbam et al. (2023).

### 2.2 Soil respiration measurements

In this study, manually measured soil respiration represents the sum of $R_A$ and $R_H$ with the respiration of ground and field layer vegetation (low enough to fit inside the measurement chamber) also included, and it is denoted with $R_{GF}$. Manual chamber measurements of $R_{GF}$ were conducted weekly during the main growing season (May-Sep) in 2020-2022. The measurement



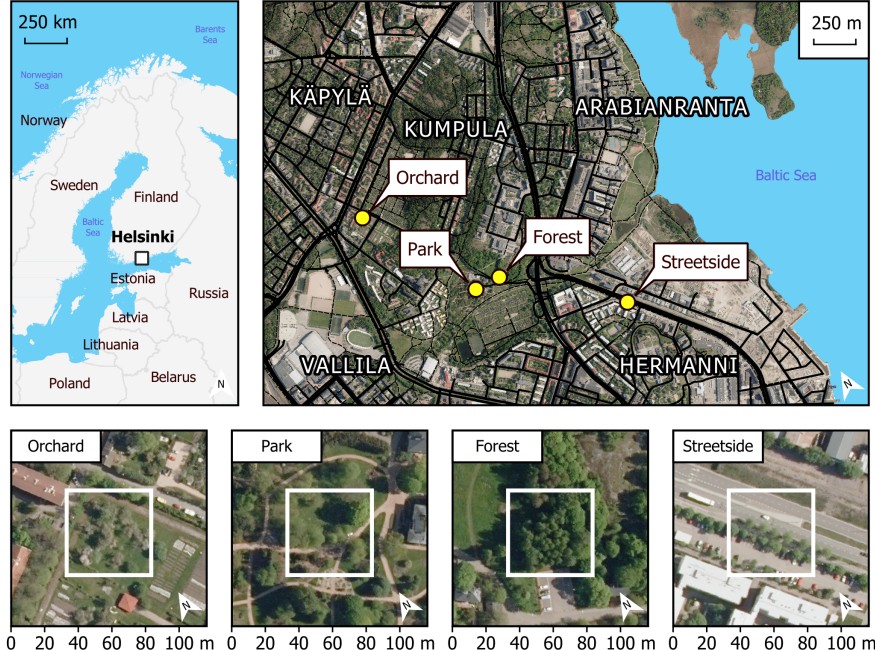

**Figure 1.** Four measurement sites (Orchard, Park, Forest, and Streetside) were located in the Kumpula and Hermanni districts in Helsinki (Finland). Site-specific panels (lower row) are scaled so the surroundings of each measurement site can be seen, while the white squares represent the more immediate locations where the manual measurements were conducted. Maps were built with the topographic database of the National Land Survey of Finland (2023), global administrative borders from GADM (2023), and orthophotos by the National Land Survey of Finland (2020).

setup consisted of a small cylindrical opaque steady-state chamber ($V = 0.007434$ m$^3$) equipped with an infrared $CO_2$ probe (GMP343, Vaisala Oyj, Vantaa, Finland), relative humidity and air temperature sensor (HMP75, Vaisala Oyj), and a battery-
powered fan to ensure air mixing within the chamber. Measurement data from the sensors were stored on site in a hand-held data logger (MI70, Vaisala Oyj). On each measurement day all sites were measured between 8 AM and 4 PM. All measurement sites were not active in all study years; a detailed overview of the measurement schedule and some exceptions to the standard protocol are described in Figure B1.

Eight chamber measurement points were systematically selected at each measurement site and the measurements were
always performed at these fixed points. A steel base frame for the chamber was installed at each point at two of the sites (Forest, Park), whereas mobile base frames were used at the other sites (Orchard, Streetside), because permanent installations would have prohibited regular activities (e.g. lawn mowing, recreational use) at the sites. The base frames were gently inserted 0.5-2 cm into the soil in order to avoid damaging the vegetation while still allowing for an airtight seal. After insertion the height of the mobile base frame was measured to determine the total chamber headspace volume needed to calculate the flux.
The heights of the permanent base frames were monitored and re-measured at least a few times each year. The closure time





of a single chamber measurement varied between 4-5 minutes, and the chamber was well ventilated between measurements. Data quality was monitored visually on-site by observing the increasing trend of $CO_2$ concentration within the chamber, and the measurements were repeated if the quality was deemed insufficient.

### 2.3 Ancillary measurements

Soil temperature at each chamber measurement point was measured (at 10 cm depth) during the chamber measurement with a hand-held soil thermometer (Pt100 and HH376, Omega Engineering Inc., Connecticut, USA). Soil moisture was measured (at 10, 20, 30 and 40 cm depths) with a soil profile probe (PR2, Delta-T Devices, Cambridge, UK) concurrently with the chamber measurements (see Figure B1 for more details). Six fibreglass access tubes (ATS1, Delta-T Devices) were installed at each site. Three readings were obtained from each tube while horizontally rotating the profile probe 120 degrees in between to ensure

spatial representativeness to all directions (Delta-T Devices Ltd., 2016). Data were stored on site in a hand-held data logger (HH2, Delta-T Devices). During the campaign years a number of access tubes at Streetside broke down due to management and construction with heavy machinery. As a result, new tubes were installed to replace the broken ones. However, this led to some variation in the number of tubes measured each week.

Soil moisture readings were first averaged separately for each depth and over each access tube. The tubes at each site were

then compared against each other, and anomalous single readings were discarded (total of 4: one at Forest and three at Park). If a tube constantly provided data that was notably different to the others, all readings from that tube were discarded (total of 2: both at Streetside).

### 2.4 Soil sampling, analysis and stock calculation

Three types of soil samples were collected from all sites at some point during the campaign years. Particle size distribution,

soil pH, and concentrations of various nutrients were analysed at a commercial lab (Eurofins Viljavuuspalvelu Oy, Mikkeli, Finland). 1 L of soil was collected at Forest, Garden, and Streetside by pooling together 16-18 individual soil core samples collected from 0–30 cm depth with a thin auger ($d$ = 2.3 cm). At Orchard 4 individual soil core samples were collected with a larger auger ($d$ = 5.0 cm). The particle size distribution was analysed according to Elonen (1971).

Samples for soil density were collected by inserting a steel cylinder ($V$ = 0.151 dm$^3$ at Orchard, $V$ = 0.2 dm$^3$ at the other

sites) horizontally into an undisturbed soil profile at 10 cm depth. The fully inserted cylinder was gently detached with the sampled soil inside it to achieve volumetric accuracy. The samples were dried at 105 °C for 48 h and the dry weights were weighed. Soil density was then calculated by dividing the sample dry weight with the cylinder volume. Five individual samples were collected at Streetside, Park, and Forest, and three samples at Orchard.

Six individual soil core samples were collected from 0–30 cm depth at each site with a soil auger ($d$ = 1.7 cm at Orchard, $d$

= 2.3 cm at the other sites) to analyse SOC and soil organic nitrogen (SON) content. The samples were sieved with 2 mm mesh sieve and dried at 105 °C for 24 h, after which the dry weights of the smaller and larger grain size classes were weighed. The samples from Orchard were, however, sieved only after drying. Total soil SOC and SON contents were determined from the dried and milled samples of soil with grain size smaller than 2 mm with an elemental CN analyser (LECO, Michigan, USA).



The results were adjusted based on the site-specifically averaged proportion of soil with grain size larger than 2 mm assuming
its SOC and SON content to be zero. Consequently, SOC and SON stocks (for 0–30 cm depth) were calculated utilising the
averaged soil density at each site.

## 2.5   Flux data processing

$CO_2$ concentration measured with Vaisala GMP343 is dependent on air pressure, air temperature, relative humidity (RH), and
oxygen ($O_2$) concentration (Vaisala, 2007). We used the automatic compensation procedures of the MI70 software to compen-
sate for the effect of air temperature and RH by utilising real time air temperature data from GMP343's internal temperature
sensor and RH data from HMP75 sensor attached to the chamber measurement setup. We checked the prevailing air pressure at
the Kumpula weather observation station (Finnish Meteorological Institute, 2023) operated by Finnish Meteorological Institute
(FMI) (N60°12'14.0", E24°57'38.9"; located 200-1000 m from the measurement sites) in the beginning of each measurement
day and used that as an input for the automatic air pressure compensation for all measurements conducted during the day. 21.0
% was used as a constant for the $O_2$ concentration compensation for all measurements.

The first 30 seconds of data were truncated from the beginning of each measurement in order to allow the chamber headspace
air to stabilise after closing the chamber. Then, the soil respiration $CO_2$ flux ($R_{GF}$) was calculated with Equation 1:

$$R_{GF} = \left( \frac{\delta C(t)}{\delta t} \right)_{t=0} \times \frac{M \times P \times V}{R \times T \times A}, \tag{1}$$

in which $\left( \frac{\delta C(t)}{\delta t} \right)_{t=0}$ is the time derivative ($CO_2$ ppm s$^{-1}$) of a linear regression during a single chamber closure, $M$ is
the molecular mass of $CO_2$ (44.01 g mol$^{-1}$), $P$ is the ambient air pressure during each measurement day (Pa), $V$ is the total
system (chamber + collar) volume (m$^3$), $R$ is the universal gas constant (8.31446 J mol$^{-1}$ K$^{-1}$), $T$ is the mean temperature
(K) inside the chamber during the closure, and $A$ is the basal area (m$^2$) of the chamber. The fits of all linear regressions were
visually inspected, and the start and end points were adjusted if the fit quality was insufficient. If the adjustments did not lead
to an acceptably linear fit, or the eventual measurement duration after the adjustments would have been less than 2 min, the
measurement was discarded.

## 2.6   Statistical analyses

To analyse for differences in $R_{GF}$, soil temperature, and soil moisture between the sites on a weekly level, Kruskal-Wallis
rank sum test (e.g. Hollander and Wolfe, 1973) was performed separately for each week's data (all years separately). When the
resulting p-value was statistically significant ($p<0.05$), pairwise Wilcoxon rank sum test was used as a post-hoc test to identify
the site pairs with statistically significant ($p<0.05$) differences. Benjamini-Hochberg method (Benjamini and Hochberg, 1995)
was utilised to correct for multiple testing while performing the Wilcoxon rank sum tests. Non-parametric tests were used
because of the non-normal distributions of the studied variables. The dataset used in the weekly analysis is depicted with
green, blue, yellow and grey in Figure B1. Kruskal-Wallis and Wilcoxon rank sum tests were also used to test for differences
between the sites in terms of soil density, SOC and SON content, and SOC and SON stock utilising the soil sample data.





After the analyses on a weekly level, linear mixed-effects (LME) models were used to analyse for differences between the sites when all years and weeks were pooled together. For the purposes of this analysis, the data were filtered to include only 1) the $R_{GF}$ measurements that had concurrent soil temperature and soil moisture data, and 2) the days when all intended sites had been measured during the same day. This dataset included a total of 1473 chamber measurements and is depicted with green and grey in Figure B1. $R_{GF}$, soil temperature, and soil moisture data were log-transformed before model building to enhance
normality.

Separate LME models were built for $R_{GF}$, soil temperature, and soil moisture: all of them had site ID as a fixed effect, and a week number and measurement point ID (access tube ID in the case of soil moisture data) as random effects (intercept) to account for the temporal (i.e. seasonal cycle, see Figure C1a) and spatial (i.e. repeated measurements at the same measurement points) hierarchies in the field design, respectively. The month number was also tested as a random effect but using the week
number improved the model performance according to Akaike's information criterion (AIC). Including the year as a random effect was also tested, but it was left out of the final model structure as it did not improve the model performance according to AIC. All models were fitted with restricted maximum likelihood (REML). Normality of model residuals was inspected with quantile-quantile (Q-Q) plots, and model quality ensured with conditional R-squared. After building the models, estimated marginal means (EMMs) were computed for each site to allow for pairwise comparison. All data analyses were conducted in
R (R Core Team, 2023) v. 4.1.1–4.2.3 utilizing the packages lme4 (Bates et al., 2015), multcompView (Graves et al., 2019), emmeans (Lenth et al., 2023), and MuMIn (Bartoń, 2023).

## 2.7   Ecosystem modelling

Daily $R_H$ at Forest and Park was modelled with the process-based land surface model JSBACH (Reick et al., 2013), the land component in the Earth system model MPI-ESM of the Max-Planck Institute for Meteorology (Giorgetta et al., 2013). The
model was driven with hourly observation-based data of air temperature, precipitation, shortwave and longwave radiation, relative humidity, and wind speed. Observations from the FMI Kumpula weather station (Finnish Meteorological Institute, 2023) were gapfilled with observations from the closely co-located urban measurement station SMEAR III (Järvi et al., 2009). Hourly ERA5-Land data (Muñoz-Sabater et al., 2021) was used to fill the remaining gaps. The gapfilled data were prepared for the period 2005–2022. In addition, ERA5-Land data were used from 1951 to 2004. Driver data prior to 1951 were randomly
generated from the period 1951–1980. The detailed simulations with modified forcing data were made for the years 2020–2022. The simulations had a common spin-up period, which included 8000 years of soil carbon spin-up. The forcing data were modified as described below.

The effect of varying UHI strength was emulated by adding up to 2.0 °C to the observed air temperature in 0.5 °C increments. According to an air temperature measurement campaign around the Helsinki urban area in 2009-2010, 2.0 °C is a realistic
premise for within-city air temperature variation as a result of UHI (Drebs, 2011). To emulate the effect of irrigation, an algorithm was created to increase the precipitation driver data based on the following criteria. Irrigation was applied from May to Sep, and the amount of water used for irrigation was estimated from summertime water consumption data obtained from the Kumpula botanical garden for 2019-2022. The need for irrigation was estimated based on both temperature and precipitation.





We used two week averages; if either the average temperature over two weeks was above 19 °C, or if the average precipitation
was below 1.4 mm/day (∼20 mm over two weeks), we added 1.7 mm/day irrigation as precipitation in the forcing data. When
both conditions were met, irrigation was increased to 5.0 mm/day. This setup resulted in similar year-to-year variation in the
emulated irrigation as what was seen in the water consumption data. In addition, a reference simulation was made using the
observation based forcing data, giving in total 6 simulations for each measurement site. All simulations for each site included
the same spin-up period for accumulating the soil carbon pools.

In JSBACH, the vegetation is represented by plant functional types (PFTs). The model was set up for simulating both
sites using the PFT representing temperate broadleaf deciduous trees. Phenology is described in JSBACH with the Logistic
Growth Phenology (LoGro-P) model (Böttcher et al., 2016), where the temporal development of the leaf area index (LAI) of
summer greens depends on temperature. The maximum LAI for each site was set based on Sentinel-2 data (Nevalainen et al.,
2022; Nevalainen, 2022), and the seasonal LAI dynamics driven by temperature were simulated by the model. In addition, the
phenology model parameters were adjusted to match the bud burst date, estimated from the Sentinel-2 data.

Soil texture classes for each site were determined based on the soil particle size distribution analyzed from the soil samples
collected at the sites. Accordingly, the parameters describing the soil properties follow the recommendations by Hagemann
and Stacke (2014), with the exception for the volumetric field capacity and wilting point, which were adjusted based on the
soil moisture measurements at each site. The root depth of the forest and park sites were set to 0.65 m and 0.45 m, respectively.

The description of the dynamics of litter and soil carbon in JSBACH is based on the Yasso07 model (Tuomi et al., 2009, 2011).
The model has five carbon pools based on the chemical quality of the organic matter: i) acid hydrolyzable, ii) water soluble,
iii) ethanol soluble, iv) non-soluble/hydrolyzable, and v) humus. Pools i)–iv) are the so-called AWEN pools. In addition, the
model keeps track of the woody and non-woody organic material, the difference between which is only the size of the litter
elements. The AWEN pools are further divided into above- and belowground pools. This results in 18 carbon pools altogether.
The carbon pools gain carbon from the litter flux from vegetation. Decomposition of the litter pools causes carbon to transfer
both to other pools and to the atmosphere, that is, as $R_H$. Each pool has a fixed loss rate determined at 0 °C with unlimited
soil water. These loss rates are then modified based on temperature, water availability, and size of the litter elements. The
temperature and water availability are described by two week averages of air temperature and precipitation, which to some
extent represent the soil temperature and moisture. The soil temperature and moisture simulated by JSBACH are not used to
calculate the $R_H$.

## 3 Results

### 3.1 Measured soil properties

Mean SOC contents (± standard deviation) at Forest, Orchard, Park, and Streetside were 3.4 % (±0.3), 2.5 % (±0.5), 3.3
% (±1.0), and 2.6 % (±0.6), respectively (Figure 2a). The corresponding mean SON contents were 0.29 % (±0.02), 0.20 %
(±0.03), 0.22 % (±0.05), and 0.13 % (±0.05) (Figure 2b) resulting into C/N-ratios of 11.9, 12.4, 15.0, and 20.1, respectively.
Mean SOC stocks (in kg m$^{-2}$) calculated for Forest, Orchard, Park, and Streetside were 10.9 (±1.0), 8.0 (±1.5), 8.6 (±2.6),





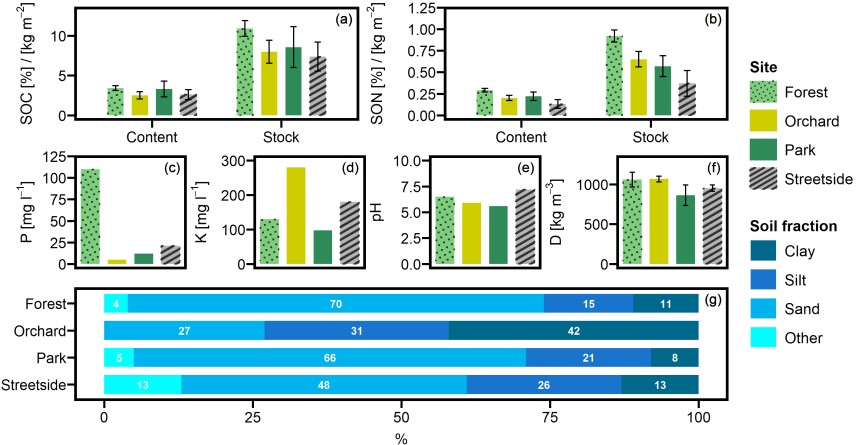

**Figure 2.** Soil a) organic carbon (SOC) content and stock, b) organic nitrogen (SON) content and stock, c) phosphorus (P) content, d) potassium (K) content, e) pH, f) density (D), and g) particle size distribution were analysed from soil samples collected at the measurement sites (Forest, Orchard, Park, and Streetside). P, K, pH and particle size distribution were analysed at a commercial lab. Grain size classes for sand, silt, and clay were 60–2000 $\mu$m, 2–60 $\mu$m, and <2 $\mu$m, respectively, and fraction "Other" refers to grain size larger than 2000 $\mu$m. Error bars denote standard deviation originating from multiple individual samples - if no error bars are shown, data originates from a pooled sample.

and 7.4 ($\pm$1.8), respectively. The corresponding mean SON stocks (in kg m$^{-2}$) were 0.92 ($\pm$0.07), 0.65 ($\pm$0.09), 0.57 ($\pm$0.12), and 0.37 ($\pm$0.15).

Overall, both SOC and SON stocks were largest at Forest and lowest at Streetside, with Orchard and Park situated in between
and somewhat on the same level (Figure 2ab). The same pattern was also visible in the SOC and SON contents, although to a less pronounced degree. In terms of statistical significance (Table 1), SOC stock at Forest was significantly ($p<0.05$) larger than at Orchard and Streetside, but there was no significant difference between Forest and Park. SON stock was significantly ($p<0.05$) largest at Forest and also significantly ($p<0.05$) larger at Orchard than at Streetside.

Soil phosphorus (P) content was distinctively higher at Forest compared to the other sites, and similarly, potassium (K)
content peaked at Orchard in comparison to the others (Figure 2cf). Differences in soil pH were less drastic, with Streetside having the highest and Park the lowest values (Figure 2e). Soil density was lowest at Park and highest at Forest and Orchard, while Streetside was situated in between the two extremes; although the differences between the extremes were small and statistically non-significant in pairwise comparison (Figure 2f, Table 1). The particle size distribution at Orchard was notably different from the other sites as the share of clay reached 42 % and there were no particles with a grain size larger than 2000
$\mu$m (Figure 2g). Consequently, when soil texture classes were determined for the sites according to the USDA classification (United States Department of Agriculture, 2017), Orchard was classified as clay whereas the other sites were classified as sandy loam.





**Table 1.** Kruskal-Wallis test and Wilcoxon rank sum test results for differences in soil density, soil organic carbon (SOC) content and stock, and soil organic nitrogen (SON) content and stock between the measurement sites (Forest, Orchard, Park, and Streetside). First, Kruskal-Wallis test was performed to detect whether there were statistically significant (p<0.05) differences, after which Wilcoxon rank sum test was utilised for pairwise comparison between the sites. Two significance levels (p<0.05 and p<0.10) were utilised regarding the latter test, and statistically significant differences between the sites are denoted with letters A–C.

| | | | | | | | | | | |
|---|---|---|---|---|---|---|---|---|---|---|
| | | | | | | Wilcoxon rank sum test | | | | |
| | Kruskal-Wallis test | | | p<0.05 | | | | p<0.10 | | |
| Variable | H statistic | P-value | Forest | Orchard | Park | Streetside | Forest | Orchard | Park | Streetside |
| Soil density | 7.97 | 0.04 | A | A | A | A | A | A | A | A |
| SOC content | 7.98 | 0.04 | A | A | A | A | B | A | AB | A |
| SOC stock | 8.02 | 0.04 | B | A | AB | A | B | A | AB | A |
| SON content | 15.2 | 0.002 | B | A | AB | A | B | A | A | C |
| SON stock | 16.8 | 0.0008 | B | A | AC | C | B | A | AC | C |

## 3.2 Measured $R_{GF}$, soil temperature, and soil moisture dynamics

Seasonal cycles were clearly visible in all of the three manually measured variables (Figures 3, D1, and D2). $R_{GF}$ and soil
temperature increased until July after which they started slowly decreasing towards autumn, and this pattern was rather similar in all study years. Soil moisture was generally at its highest in May and September and followed the precipitation events during the summer months. Its seasonal cycle had the most year-to-year variation as a result of varying precipitation regimes during the study years. For instance, there was a distinct local heatwave and drought in Helsinki during summer 2021 (see Ahongshangbam et al., 2023), which can also be seen in the decreasing trend in the measured soil moisture during June and
July (Figure 3). After the drought a peak in $R_{GF}$ was observed in the measurements of week 30.

## 3.3 Differences in $R_{GF}$, soil temperature, and soil moisture between the sites

When considering the measurement data on a weekly level, the percentage of weeks (2020-2022 combined) that featured at least one statistically significant (p<0.05) difference between the sites in terms of either $R_{GF}$, soil temperature, or soil moisture were 33 %, 83 %, and 36 %, respectively. Thus, soil temperature is clearly the variable with the highest number of observed
differences between the sites during our study period. Most commonly, Streetside differed from the others when data were available from there (2020–2021) but significantly higher momentary temperatures were recorded also in Orchard compared to the sites with higher tree cover density (Park and Forest). The differences occurred continuously throughout the study period, whereas the differences in $R_{GF}$ and soil moisture were occurring less regularly, being perhaps slightly centred around the beginning and the end of the growing season, at least in 2021 and 2022 (Figures 3 and D2). There also did not seem to be
any clear causation of significant differences in soil temperature or moisture triggering significant differences in $R_{GF}$, as i) the





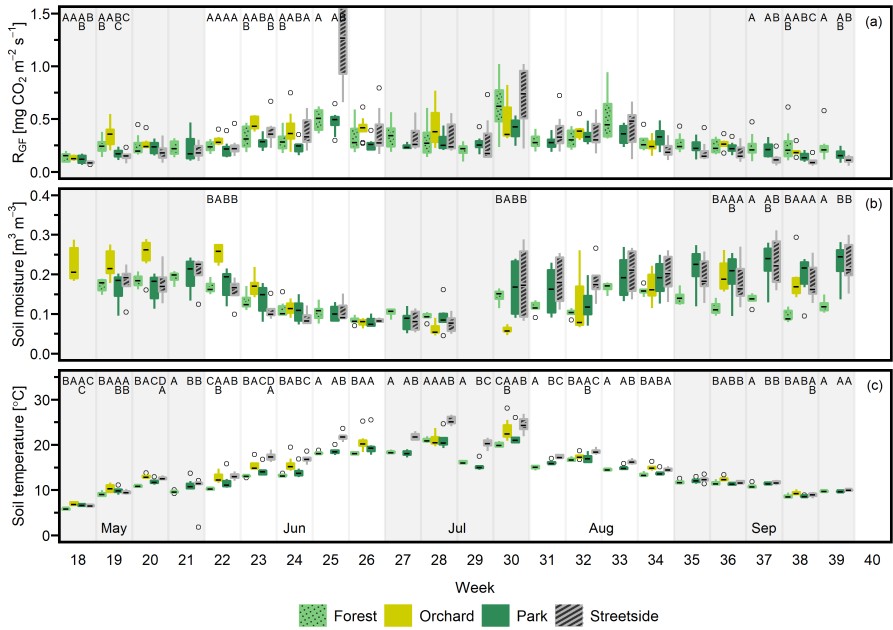

**Figure 3.** a) Soil respiration ($R_{GF}$), b) soil moisture (at 10 cm depth), and c) soil temperature (at 10 cm depth) were measured weekly at four measurement sites (Forest, Orchard, Park, and Streetside) in 2021. Here, boxes are arranged chronologically by week number, and the sites are always presented in the order that is shown in the legend. Background shading indicates the month. Empty circles are outliers. Letters A–D denote statistically significant (p<0.05) differences between the sites.

more infrequently occurring differences in $R_{GF}$ and soil moisture did not necessarily co-occur, and ii) most of the weeks that featured significant differences in soil temperature did not feature differences in $R_{GF}$.

LME models (Table 2) were used to calculate the EMMs of the measured variables for each of the sites utilising the whole dataset from 2020-2022. EMMs of $R_{GF}$ (in mg $CO_2$ m$^{-2}$ s$^{-1}$) for Forest, Orchard, Park, and Streetside were 0.270, 0.273,

0.242, and 0.242, respectively, and there were no statistically significant (p<0.05) differences between the sites (Figure 4a). EMMs of soil temperature (in °C) for Forest, Orchard, Park, and Streetside were 12.8, 13.7, 13.2, and 14.7, respectively (Figure 4b). According to pairwise comparisons, Streetside was statistically significantly (p<0.05) the warmest measurement site, Orchard was significantly warmer than Forest, and there were no significant differences between Forest and Park or Park and Orchard. EMMs of soil moisture (in m$^3$ m$^{-3}$) for Forest, Orchard, Park, and Streetside were 0.137, 0.137, 0.150, and

0.164, respectively (Figure 4c), and there were no statistically significant (p<0.05) differences between the sites. Regarding the random effects featured in the LME models, measurement point ID explained 24 %, 1 %, and 12 % of the leftover variance (i.e. after the fixed effects were considered) for the models of $R_{GF}$, soil temperature, and soil moisture, respectively, while week number correspondingly explained 30 %, 83 %, and 36 % (Table 2).





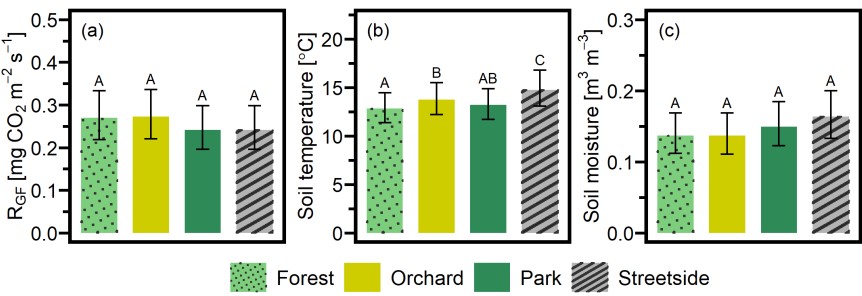

Forest  Orchard  Park  Streetside

**Figure 4.** Three separate linear mixed-effects (LME) models were built to study the differences in a) soil respiration ($R_{GF}$), b) soil temperature (at 10 cm depth), and c) soil moisture (at 10 cm depth) between the four measurement sites (Forest, Orchard, Park, and Streetside). Estimated marginal means (EMMs) were computed for each variable at each site and the statistically significant ($p<0.05$) differences between the sites are reported with letters A-C. Error bars denote 95 % confidence interval.

**Table 2.** Details of the three separate linear mixed-effects (LME) models that were built to assess the differences in soil respiration ($R_{GF}$), soil temperature (at 10cm depth), and soil moisture (at 10cm depth) between the four measurement sites (Forest, Orchard, Park, and Streetside).

| | Fixed effects[a] | | | | Random effects[b] | | | | |
|---|---|---|---|---|---|---|---|---|---|
| Response variable | Orchard | Forest | Park | Streetside | Point ID | Week | Residual | AIC | $R^2$ (cond.) |
| Soil respiration | 0.272 | 0.270 | 0.241 | 0.241 | 0.059 | 0.075 | 0.111 | 1149.9 | 0.55 |
| [mg $CO_2$ m$^{-2}$ s$^{-1}$] | (0.11) | (0.12) | (0.12) | (0.12) | | | | | |
| Soil temperature | 13.71 | 12.78 | 13.23 | 14.78 | 0.00096 | 0.076 | 0.014 | -1867.1 | 0.85 |
| [°C] | (0.059) | (0.018) | (0.018) | (0.019) | | | | | |
| Soil moisture | 0.137 | 0.137 | 0.150 | 0.163 | 0.033 | 0.099 | 0.142 | 1107.6 | 0.49 |
| [m$^3$ m$^{-3}$] | (0.10) | (0.11) | (0.11) | (0.11) | | | | | |

[a] Fixed effects are reported as estimate (standard error). [b] Variance explained by the two random effects included in the models, and the residual variance after the random effects were considered.

## 3.4 Modelled $R_H$ dynamics

Overall, the modelled $R_H$ was considerably smaller (approximately 50 %) than observed $R_{GF}$, but showed similar seasonal dynamics as $R_{GF}$ in irrigated Park and non-irrigated Forest with a few exceptions (Figure 5). First, observations included short peaks of high emissions after a rapid increase in soil moisture, especially in 2021, which were not predicted by the model. Second, the observed $R_{GF}$ did not decrease like the non-irrigated (i.e. reference simulation) $R_H$ in Forest in early 2022, but instead $R_{GF}$ increased like $R_H$ in the irrigated simulations before again following the non-irrigated $R_H$ in the second half of

the season (Figure 5c). Lastly, the observed $R_{GF}$ in Park did not increase like the irrigated $R_H$ predicted nor decrease like the non-irrigated $R_H$ during July 2021 but stayed rather stable (Figure 5e). Also, from mid-May to late August 2022, $R_{GF}$ in Park was mostly quite stable unlike the modelled dynamics (Figure 5f).




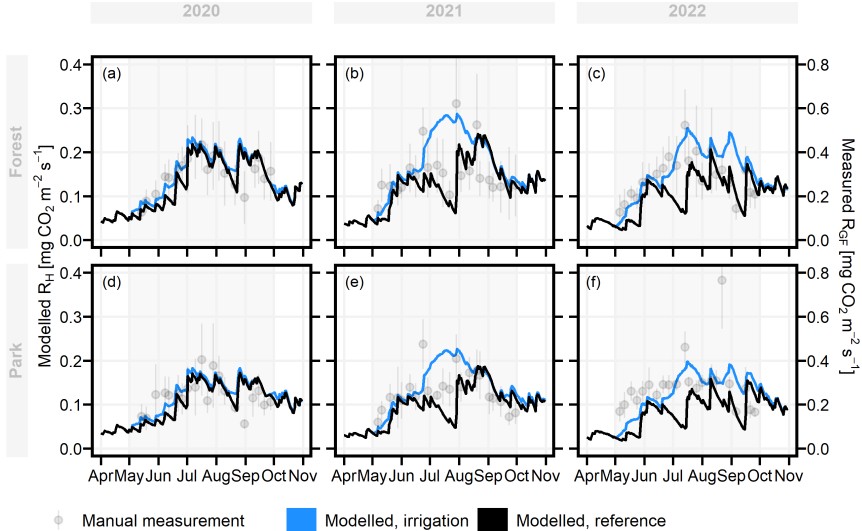

**Figure 5.** JSBACH modelled daily heterotrophic soil respiration ($R_H$, left axis) (both reference and irrigation simulation) showed similar temporal dynamics in comparison with the manually measured soil respiration ($R_{GF}$, right axis). Manual measurements are portrayed as mean ± standard deviation, and background shading indicates the study period May–Sep.

### 3.5 Modelled impact of UHI and irrigation on $R_H$

The modelled effect of elevated air temperature on $R_H$ varied only slightly between the two measurement sites (Table 3, Figure 325 6). When the daily mean momentary $R_H$ fluxes were summed over the study period of May-Sep separately for each year (Table 3), an increase of 0.5 °C in air temperature increased $R_H$ on average by 2.0 % and 1.3 % at Forest and Park, respectively. Based on the averaged results, an increase of 2 °C in air temperature within a city, as a result of the UHI, would result in 6.6-8.0 % increase in local $R_H$ $CO_2$ emissions depending on the green space type.

Simulated irrigation had a major effect in increasing $R_H$ during the dry summers of 2021 and 2022, during which the relative 330 increase in the cumulative $R_H$ $CO_2$ emissions over the study period May-Sep was in the range of 37.0-38.0 % and 52.3-52.7 %, respectively (Table 3, Figure 6). Again, the effect was considerably similar for both measurement sites. As the weather during the 2020 study period was more typical for Helsinki, the effect of irrigation was less pronounced (10.9-11.1 %), but even then the increase in $R_H$ was more than what was seen with the elevated air temperatures.

## 4 Discussion

Quantifying the biogenic C stocks and C uptake potential in urban green spaces both now and in the future requires not only aboveground C stock estimates but also an understanding of the soil and the C emissions arising from it. In this study, we collected data on $R_{GF}$ and its main environmental drivers at four measurement sites representing different types of tree-covered





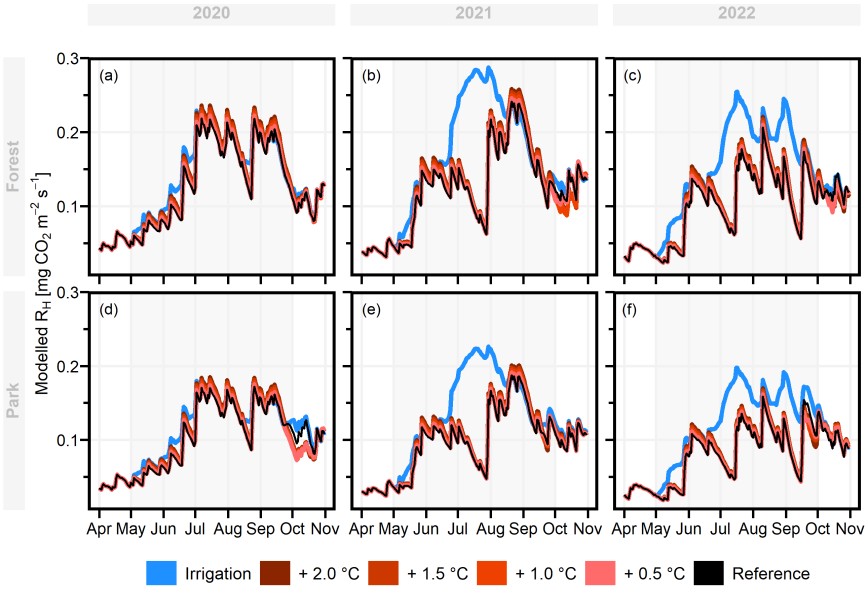

**Figure 6.** Daily heterotrophic soil respiration ($R_H$) at Forest and Park was modelled with JSBACH to study the effect of the urban heat island (UHI) and irrigation. During the study period of May-Sep (indicated with background shading), air temperature was increased by 0.5, 1.0, 1.5, and 2.0 °C, and an irrigation algorithm was used to simulate lawn irrigation during dry periods. A reference simulation was conducted separately for both measurement sites (Forest and Park) with the observed local weather conditions of each year.

**Table 3.** Daily heterotrophic soil respiration ($R_H$) at Forest and Park was modelled with JSBACH under varying environmental driver simulations. The daily carbon dioxide ($CO_2$) emissions were summed over the study period of May-Sep and compared to a reference run conducted with the observed local weather conditions of each year: positive values presented in the table imply an increase (in %) in $R_H$ compared to the reference simulation. Results from the study years are shown both individually and as a mean of all study years.

|  |  | 2020 |  | 2021 |  | 2022 |  | Mean |  |
|---|---|---|---|---|---|---|---|---|---|
|  |  | Forest | Park | Forest | Park | Forest | Park | Forest | Park |
| Temperature | + 0.5 | 2.1 | 1.3 | 1.8 | 1.8 | 2.1 | 0.6 | 2.0 | 1.3 |
|  | + 1.0 | 4.2 | 3.4 | 3.7 | 3.5 | 4.1 | 2.0 | 4.0 | 3.0 |
|  | + 1.5 | 6.2 | 5.4 | 5.7 | 5.2 | 6.2 | 3.7 | 6.0 | 4.9 |
|  | + 2.0 | 8.2 | 7.5 | 7.6 | 6.8 | 8.2 | 5.3 | 8.0 | 6.6 |
|  | Irrigation | 10.9 | 11.1 | 37.0 | 38.0 | 52.3 | 52.7 | 31.6 | 32.2 |

urban green spaces expecting the drivers and, consequently, the resulting $R_{GF}$ to differ among the sites. However, despite evident differences in management practices and standing tree volume as well as in observed SOC and soil temperature, the





observed $R_{GF}$ was equal between the sites, except for momentary occasions. In addition, the impact of the UHI on $R_H$ was minor when compared to the effect of irrigation.

     Overall, the estimated marginal means (EMMs) of $R_{GF}$ at the different green space types in May-Sep presented in the current study (Figure 4) are of a similar order of magnitude (approximately 0.2-0.3 mg $CO_2$ m$^{-2}$ s$^{-1}$) to $R_S$ measured in urban green spaces (forest, lawn, landscaped) in Boston (Decina et al., 2016; Garvey et al., 2022), and under coniferous and

deciduous trees in a botanical garden in Moscow (Goncharova et al., 2018). Wu et al. (2016) measured $R_S$ specifically at the boundary between green space and impervious surface in Beijing; the mean momentary $R_S$ values there were notably high especially right at the impervious surface border, but in many cases decreased to a rather similar magnitude with our results when moving more than 1.5 meters away from the border. The highest mean $R_S$ rate they reported was 0.85 mg $CO_2$ m$^{-2}$ s$^{-1}$, which is something that was reached (and even surpassed) in our data during singular measurement weeks, but not in seasonal

means.

     The currently measured urban $R_{GF}$ rates were notably higher than some $R_S$ rates measured in non-urban ecosystems in Southern Finland such as barley fields (on average 0.10–0.14 mg $CO_2$ m$^{-2}$ s$^{-1}$ Koizumi et al. (1999)) or forestry-drained peatlands ($R_H$ only, on average 0.08-0.10 mg $CO_2$ m$^{-2}$ s$^{-1}$ Minkkinen et al. (2007)). In contrast, summertime forest floor $R_S$ rates reported in southern (approximately 0.17-0.33 mg $CO_2$ m$^{-2}$ s$^{-1}$ (Ryhti et al., 2022)) and northern (approximately

0.23-0.35 mg $CO_2$ m$^{-2}$ s$^{-1}$ (Kulmala et al., 2019)) Finland were only slightly lower than our seasonal EMMs, although our weekly $R_{GF}$ rates during the peak summer months Jun-Aug tended to frequently exceed the range of the non-urban forest floor $R_S$. Similarly, summertime agricultural $R_S$ rates reported by Heimsch et al. (2021) range, on average, between 0.23-0.35 mg $CO_2$ m$^{-2}$ s$^{-1}$. As our study lacks non-urban measurements to act as points of reference, we cannot reach such a clear conclusion of $R_S$ in urban ecosystems being more than twofold in magnitude compared to their non-urban counterparts as

concluded by Decina et al. (2016), even though our results do support the premise of elevated $R_S$ in urban areas.

     Our measurements of SOC stocks in urban green space (on average 7.37-10.92 kg m$^{-2}$) are in line with previous research, although being situated rather towards the lower end of a broad spectrum (Table 4). However, it needs to be noted that the sampling depth in many of the previous studies has extended down to 100 cm, whereas our SOC samples represent the top 0-30 cm layer, which makes straightforward comparison difficult. Nevertheless, the stocks are still comparably or even notably

higher than what has been measured in Finnish non-urban ecosystems, for example, in Scots pine (*Pinus sylvestris* L.) and Norway spruce (*Picea abies* (L.) H. Karst) dominated forest plots throughout Finland (on average 5.49 and 8.32 kg m$^{-2}$, respectively, considering both organic layer and mineral soil (Lindroos et al., 2022)), and in agricultural lands in Finland (on average 4.1-6.7 kg m$^{-2}$, for 0-15 cm depth (Heikkinen et al., 2013)).

     The highest SOC stock at our measurement sites was in a deciduous urban forest (Forest), where the litter C input to the

soil is undoubtedly a lot higher than at the other sites that are under a more active management regime in terms of raking and removal of fallen branches, etc. This result would support the importance of non-intensively managed and infrequently disturbed urban forests, not only for their aboveground C stocks, but especially for their SOC (see also e.g. Yesilonis and Pouyat, 2012; Lindén et al., 2020). Also, it needs to be noted that our measurement sites were of somewhat varying age (Table A1), which can have an impact on the observed SOC levels. In terms of temporal trends, urban SOC stock tends to first decrease



**Table 4.** Soil organic carbon (SOC) stocks at various urban green space types reported in previous literature, arranged in descending order.

| Reference | Location | Green space type | Sampling depth [cm] | SOC stock [kg m$^{-2}$] | |
|---|---|---|---|---|---|
| Riikonen et al. (2013) | Finland, Helsinki | Old street trees | 0–90 | ~40 | a) |
| Setälä et al. (2016) | Finland, Lahti | Park lawn | 0–50 | 22-35 | b) |
| Edmondson et al. (2014) | UK, Leicester | Urban woodlands | 0–100 | 14–26 | |
| Pouyat et al. (2006) | USA, Chicago | Residential grass | 0–100 | 16.3 | |
| Lindén et al. (2020) | Finland, Helsinki | Park lawn | 0–90 | 15.5 | g) |
| Edmondson et al. (2014) | UK, Leicester | Urban grassland | 0–100 | 15 | |
| Pouyat et al. (2006) | Russia, Moscow | Residential grass | 0–100 | 14.6 | c) |
| Pouyat et al. (2009) | USA, Baltimore | Residential grass | 0–100 | ~12.2 | |
| Lindén et al. (2020) | Finland, Helsinki | Park lawn | 0–90 | 10.4 | |
| Pouyat et al. (2006) | USA, Baltimore | Park lawn | 0–100 | 9.9 | |
| Dorendorf (2014) | Germany, Hamburg | Lawn | 0–30 | 9.7 | |
| Riikonen et al. (2017) | Finland, Helsinki | New street trees | 0–90 | 9 | d) |
| Shchepeleva et al. (2017) | Russia, Moscow | Lawn | 0–30 | ~6 | e) |
| Kaye et al. (2005) | USA, Colorado | Lawn | 0–15 | 4.7 | f) |
| Pouyat et al. (2006) | China, Hong Kong | Park lawn | 0–100 | 4.2 | c) |
| Bae and Ryu (2015) | South Korea, Seoul | Park lawn | 0–100 | 3.4 | |

a) From restricted growing media. b) Bulk density was not measured. c) Calculated based on data from an earlier study. d) Stone-based growing media. e) Rather newly-established lawn. f) Irrigated and fertilised. g) Under vegetation.

as a result of construction and possible land use change, but can subsequently increase to a level surpassing that of non-urban areas (Pataki et al., 2006). Havu et al. (2022) inspected this in their modelling study: after constructing a new streetside green space, the annual $R_S$ C emissions were high enough to supersede the amount of C sequestered annually by the newly planted street trees for the first 12-14 years after plantation. Our measurement sites mainly represent urban green spaces at such a life cycle stage in which the possible differences in SOC stock arising from the initial construction may have already levelled out

but the long-term development possibly still remains largely unseen.

Our initial hypothesis was that the overall heterogeneity typical for urban environments would be likely to establish varying levels of soil temperature and soil moisture at the four measurement sites – even though all of them were located within 2 km from each other. Indeed, soil temperature at Streetside was the highest of all measurement sites, which can most likely be explained by its surroundings: it was the site surrounded with the most extensive sealed surface cover and highest building

density (see Figure 1 and also Ahongshangbam et al. (2023)) that were both likely to contribute to a more pronounced local UHI and consequently, an elevated soil temperature. Soil temperature at Orchard was also significantly higher than at Forest, which could be explained by differences in their vegetation characteristics: at Orchard, the sparse apple trees grew on a lawn, whereas at Forest the measurement points were situated under a more closed canopy formed by distinctively taller trees, thus



being effectively surrounded and shaded by the forest itself in all cardinal points except for a small sector (i.e. forest edge) in
southwest. Therefore, Orchard was likely to receive more direct sunlight as a result of less shading from its surroundings, and
more of that sunlight would have been reaching the ground level to warm up the soil due to lower tree cover density than what
was the case at Forest.

Despite the observed significant differences in soil temperature, soil moisture levels were significantly different at the mea-
surement sites only during some individual weeks, and there was no clear pattern of some sites being significantly different
from others when analysing the dataset as a whole. Uniform soil moisture conditions could possibly be one of the prominent
reasons for the fact that no significant differences were observed in $R_{GF}$ either; according to Goncharova et al. (2018), soil
moisture is the main factor controlling urban $R_S$ during summer when soil temperature has exceeded 10 °C. Another reason
for the observed uniformity could be that SOC stocks at the sites were significantly different and the pattern was approximately
the opposite of what was observed with soil temperature: the warmest site had the lowest SOC stock and vice versa. Since
$R_S$ is partly the result of decomposing SOC stock, a lower SOC stock to begin with could possibly permit the increase in $R_S$
even with the observed elevated soil temperature. Although, drawing a rigid conclusion on such compensatory effects would
warrant a more specifically tailored measurement setup than what this present study has to offer.

In general, the results of our LME analysis were in line with the findings of the week-level analysis. On a weekly level,
soil temperature was the variable with the most frequently occurring statistically significant (p<0.05) differences between the
measurement sites, and it was also the only variable with statistically significant differences between the sites in the LME
analysis. The amount of variance explained by the random effects included in the LME models (Table 2) indicates that there
was some systematic spatial variation in $R_{GF}$ between the individual measurement points at each site, whereas there was hardly
any variation in soil temperature. Previous research has also demonstrated $R_S$ to commonly have notable spatial variation even
in small scales (see e.g. Soe and Buchmann, 2005; Martin and Bolstad, 2009). Week number was especially good in explaining
the temporal variation in soil temperature, which is likely due to it having the most pronounced seasonal cycle.

We expected a more pronounced effect of the UHI on $R_H$ than found: comparing increases in $R_H$ associated with either a
minor increase in air temperature or active irrigation revealed the latter to be much more significant to the magnitude of the
combined $R_H$ $CO_2$ emissions of the growing season. On average, increasing air temperature by 2 °C increased $R_H$ by less than
10 % compared to the reference run, whereas the increase produced by irrigation was 30 % higher – although in 2020, when
the weather during the growing season was more typical for Helsinki than in the other (i.e. dryer) study years, the increase by
irrigation was only slightly over 10 %. The surprisingly small impact of temperature on $R_H$ is supported by the small variation
in measured $R_{GF}$ between the measurement sites despite the significant temperature differences. At the same time, it must be
noted that irrigation during drought will not only increase the C emissions by stimulating $R_S$, but also improve and sustain the
livelihood of the vegetation, and thus allow for more continuous, and even increased, C sequestration that can result in a net
negative impact in the overall C balance of the ecosystem (see e.g. Wu et al., 2008; Olsson et al., 2014; Trémeau et al., 2023).
Furthermore, irrigation has been shown to also lower soil temperature (Cheung et al., 2022b, a), which hinders $R_S$. Because
of the multitude of intertwined factors determining the ultimate impact of irrigation on the C balance of an urban ecosystem, a



properly controlled empirical experiment is still needed to reach credible conclusions. To add to the comparably short temporal viewpoint of this study, addressing the long-term effects of irrigation on SOC warrants further examination.

It is tricky to compare modelled $R_H$ with the observations, which in this study also included the release of $CO_2$ during plant metabolic processes, that is, the $R_A$ of tree roots, lawn, and other ground and field layer vegetation. Accurate observations of strictly $R_H$ would enable a direct comparison but such data are difficult to collect due to the interlinked nature of the different soil processes. For example, widely used root exclusion techniques, such as trenching, suffer from increased root litter, alterations in soil moisture, and changes in the activity and composition of microbial communities (Hanson et al., 2000;

Ryhti et al., 2022). Also, the removal of belowground parts of ground vegetation, such as lawn, would affect the temperature and moisture of topsoil and thus also the heterotrophic activity. Consequently, utilising process-based models is a cost-efficient method to partition the $R_S$ components.

The share of $R_A$ in total soil respiration naturally depends on the amount of vegetation and on soil properties, such as fertility or the quality and quantity of soil organic matter (Mäki et al., 2022). Hanson et al. (2000) estimated that, on average,

root respiration contributes annually 49 % of total $R_S$ for sites with forest vegetation, based on 37 published field-based studies. However, the share might change during the summer as the seasonal dynamics of root respiration in trees are influenced by environmental factors and phenological variations (Hopkins et al., 2013; Pumpanen et al., 2015). In this study, the simulated $R_H$ was roughly 50 % of the observed $R_{GF}$ with no clear seasonal discrepancies. However, momentary changes in one might be hidden by opposite changes in the other.

As expected, the temporal patterns of the observations followed the irrigated simulation in the Park and the non-irrigated simulation in the Forest with only a few exceptions. First, the observed $R_{GF}$ in Park did not increase like the irrigated simulation in the year 2021. This difference is likely attributed to the actual irrigation scheme at that time, as the garden managers avoided watering the scientific instruments within our measurement site, resulting in somewhat less irrigation in comparison to most of the lawns within the park. Second, during the rainless period in early 2022, observed respiration in Forest did not decrease

as predicted by the simulation whereas the simulation accurately captured the subsequent reduction later in the season. This probably arises from increased autotrophic activity in the early season as it is well-known that roots are less sensitive to a decrease in topsoil moisture compared to heterotrophic activity (Ryhti et al., 2022) and that they can also acquire water from deeper soil layers. Therefore, we presume that most of the observed decreases in $R_S$ during summer periods resulted from drought-restricted heterotrophic activity.

The "Birch effect" following a rain event in forest ecosystems is a widely recognised phenomenon (Birch, 1958; Jarvis et al., 2007). We can assume that some of the observed $CO_2$ peaks after a rapid increase in soil moisture probably arose from autotrophic activities, but most likely the majority of them related to the fast breakdown of easily decomposable carbon substrates that have accumulated during the dry period. In an experimental field study, Unger et al. (2010) gained support for their hypothesis that rapid mineralization of either dead microbial biomass or osmoregulatory substances released by soil

microorganisms in response to hypo-osmotic stress is behind the phenomenon. However, Yasso soil carbon model (Tuomi et al., 2009, 2011) included in JSBACH does not include such processes even though there are indications that sequential dry periods followed by heavy rains favour the accumulation of SOC compared with management schemes that maintain the soil moisture





close to field capacity (Kpemoua et al., 2023). In the face of changing precipitation regimes and irrigation recommendations, understanding the longstanding impacts of the Birch effect and irrigation on the longevity of urban SOC in boreal regions requires further controlled experiments.

Our study presents a valuable and temporally extensive dataset of chamber measurements of soil respiration and measurements of SOC stocks in urban green spaces, both topics that are still globally lacking measurement-based data. We acknowledge that a study setup with wider spatial coverage would have been useful in giving more grounds for conclusions regarding the specific characteristics of different tree-covered urban green space types. It would have been optimal to have more replicates of each type and to have them situated over a broader spatial scale within Helsinki, perhaps utilising land use or land cover data for a more nuanced site stratification approach. Although, conducting measurements with such a spatially extensive setup would require much more resources and also hinder the frequency with which single sites could be visited compared to the temporal coverage that was achieved with the present setup. It is hard to explicitly account for the singular effects of multiple important and co-affecting environmental variables in a field measurement setup like ours, in which the main idea is to measure the studied phenomena of interest ($R_S$) in the naturally varying environmental conditions rather than conducting measurements in a strictly controlled study setup.

## 5   Conclusions

As cities are becoming increasingly interested in utilising urban vegetation and soil to sequester and store carbon, measurement data is needed to properly understand the biogenic carbon cycle in urban ecosystems. We carried out an extensive field measurement campaign on soil respiration across a variety of tree-covered urban green spaces in Helsinki to investigate whether the varying urban structure would create variation in the key drivers of soil respiration and, consequently, affect the soil respiration rates. The management practices and standing tree volume between the sites were clearly different and the soils had statistically significant differences in soil temperature as well as soil organic carbon and nitrogen stocks, but the only differences in soil respiration we could distinguish seemed momentary and sporadic. Process-based model simulations showed that the increase in heterotrophic soil respiration over the growing season caused by elevating air temperature by 2 °C, to simulate the urban heat island effect, was less than 10 %, whereas irrigation of urban green spaces created a stronger increase averaging more than 30 %, and could reach over 50 % during a drier year. The observed consistency of modelled and measured data encourages the use of process-based models in simulating the urban biogenic carbon cycle.

Overall, our findings challenged some of our initial hypotheses, and would encourage further studies on the topic, for example, utilising a measurement site setup with a broader spatial span and more site type replicates. Based on our results, different soil temperature conditions are likely not the sole explanation for the previously discussed differences in the magnitude of soil respiration between urban and non-urban ecosystems – we cautiously emphasise the role of irrigation and soil moisture and hope to motivate further studies on the topic. We would also tend to agree with Decina et al. (2016) on the roles of possible organic amendments and the soil itself, especially soil organic carbon, in generating the differences in soil respiration between urban and non-urban ecosystems. Similarly, soil characteristics are likely an important factor in establishing variation



in soil respiration within a city, but disentangling their specific effects from those of soil temperature and moisture remains a challenge.

*Data availability.* The measurement data used in the study can be accessed and downloaded at Finnish Meteorological Institute B2SHARE: http://hdl.handle.net/11304/9961c5ae-e967-4033-9bfa-3f734307def0 and https://doi.org/10.57707/fmi-b2share.f7ba414bfd3642168ac38a95835b06bc

495 (Karvinen, 2023).



# Appendix A: Measurement site details

**Table A1.** Vegetation and management characteristics at the measurement sites: main tree species, mean height (m) of the main tree species, mean diameter at breast height (DBH) (cm) of the main tree species, approximate age of the main tree species i.e. years since plantation, ground vegetation type, and the presence of irrigation, fertilisation, and mowing.

| Site ID | Main tree species | Mean height (m) | Mean DBH (cm) | Age (y) | Ground vegetation | Irrigation | Fertilisation | Mowing |
|---|---|---|---|---|---|---|---|---|
| Forest | Silver birch (*Betula pendula* Roth) | 22 | 23.6 | 35 | Forest vegetation | No | No | No |
| Orchard | Apple (*Malus domestica* Borkh.) | 6.5 | 30 | ~72 | Managed lawn | No | No | Yes |
| Park | Small-leaved linden (*Tilia cordata* Mill.) | 12.5 | 26.3 | 26 | Managed lawn | Yes | Yes | Yes |
| Streetside | Common linden (*Tilia x europaea* L.) | 10 | 19.5 | 34 | Managed lawn | No | No | Yes |





**Table A2.** Measurement site locations and soil characteristics. Values determined from multiple individual samples are given as mean (standard deviation), otherwise values represent a pooled sample. P, K, pH, and particle size distribution were analysed at a commercial lab.

| Site ID | Coordinates (WGS84) | Soil texture (USDA)[a] | Soil density [kg m$^{-3}$] | P [mg l$^{-1}$] | K [mg l$^{-1}$] | pH | Particle size[b] | | | | SOC | | SON | |
|---|---|---|---|---|---|---|---|---|---|---|---|---|---|---|
| | | | | | | | Clay [%] | Silt [%] | Sand [%] | Other [%] | Content [%] | Stock [kg m$^{-2}$] | Content [%] | Stock [kg m$^{-2}$] |
| Forest | N 60°12'07.7" E 24°57'33.0" | Sandy loam | 1060 (94) | 110 | 130 | 6.5 | 11 | 15 | 70 | 4 | 3.4 (0.31) | 10.92 (0.99) | 0.29 (0.02) | 0.92 (0.07) |
| Orchard | N 60°12'30.17" E 24°56'57.77" | Clay | 1068 (35) | 4.9 | 280 | 5.9 | 42 | 31 | 27 | 0 | 2.5 (0.46) | 7.99 (1.46) | 0.20 (0.03) | 0.65 (0.09) |
| Park | N 60°12'08.4" E 24°57'21.4" | Sandy loam | 864 (131) | 12 | 97 | 5.6 | 8 | 21 | 66 | 5 | 3.3 (0.99) | 8.57 (2.56) | 0.22 (0.05) | 0.57 (0.12) |
| Streetside | N 60°11'51.6" E 24°58'13.2" | Sandy loam | 953 (40) | 21 | 180 | 7.2 | 13 | 26 | 48 | 13 | 2.6 (0.64) | 7.37 (1.82) | 0.13 (0.05) | 0.37 (0.15) |

[a] Soil texture class according to the USDA classification (United States Department of Agriculture, 2017). [b] Grain size classes for sand, silt and clay were 60-2000 $\mu$m, 2-60 $\mu$m, and <2 $\mu$m, respectively, and fraction "Other" refers to grain size larger than 2000 $\mu$m.



## Appendix B:  Measurement dataset details

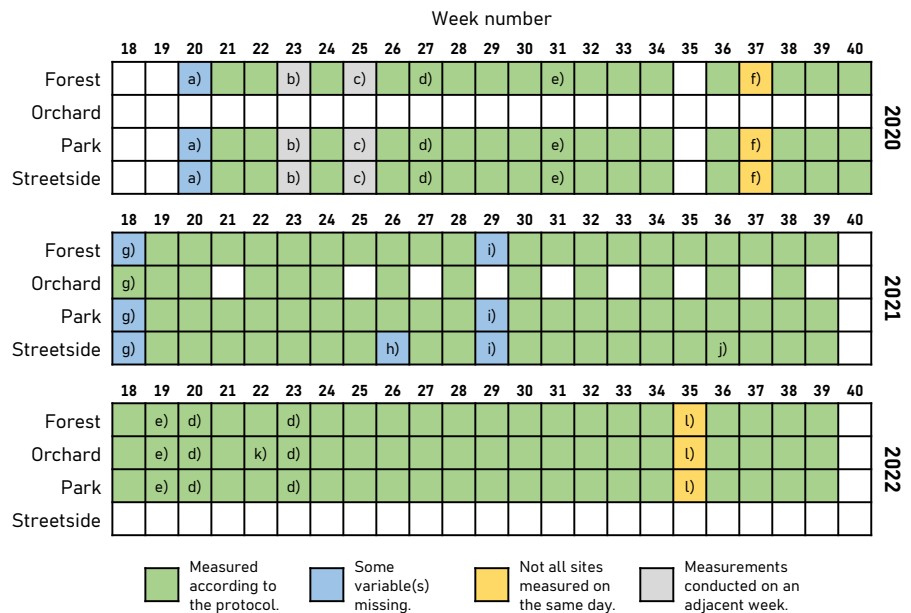

**Figure B1.** Overview of the schedule for manual soil respiration measurements and the concurrent soil temperature and soil moisture measurements. a) Soil temperature and soil moisture were not measured. b) Measurements were conducted on Monday of week 24, whereas measurements of week 24 were conducted on Friday. c) Measurements were conducted on Monday of week 26, whereas measurements of week 26 were conducted on Friday. d) All measurements were conducted in the afternoon. e) All measurements were conducted in the afternoon, after some rain in the morning. f) Park and Streetside were measured on Wednesday, whereas Forest was measured on Friday. g) Soil moisture was measured only at Orchard. h) Soil temperature was not measured at Street. i) Soil moisture was not measured. j) Soil temperature missing from three measurement plots at Street (S6-S8). k) At Orchard, only two flux measurement plots (and their respective soil temperatures) were measured. Still, all soil moisture measurements were conducted. l) Forest and Park were measured on Tuesday, whereas Orchard was measured on Wednesday.





# Appendix C: $R_{GF}$ dataset for LME model building

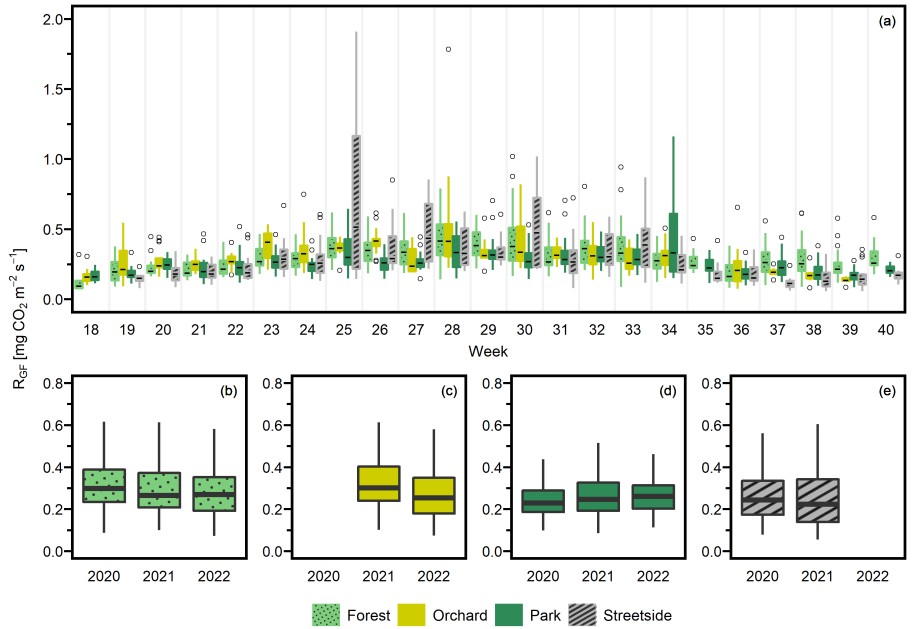

**Figure C1.** All soil respiration ($R_{GF}$) measurements that were used in building the linear mixed-effects (LME) models. a) All measurements from 2020-2022 grouped by site and arranged chronologically by week number. Sites are always presented in the same order that is shown in the legend. Outliers are marked with empty circles. b)-e) All measurements from each site pooled together separately for each year. Week number was added as a random effect in the models to account for the temporal hierarchy in the data, but year was not included, since there were no apparent differences between the three study years. Outliers are not portrayed in panels b)-e) to enhance clarity.



## Appendix D: Weekly measurements of 2020 and 2022

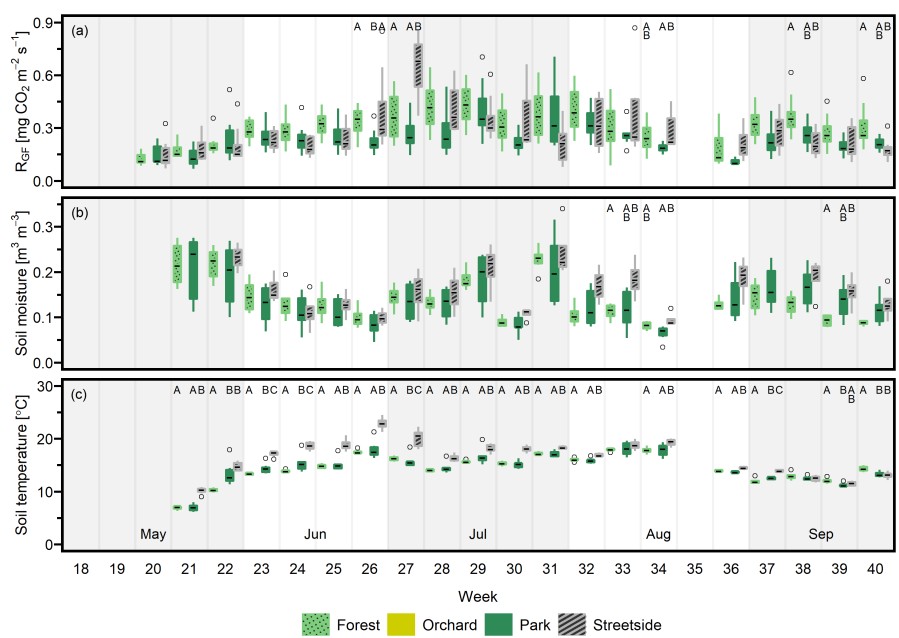

**Figure D1.** a) Soil respiration ($R_{GF}$), b) soil moisture (at 10 cm depth), and c) soil temperature (at 10 cm depth) were measured weekly at three measurement sites (Forest, Park, and Streetside) in 2020. Here, boxes are arranged chronologically by week number, and the sites are always presented in the order that is shown in the legend. Background shading indicates the month. Empty circles are outliers. Letters A-C denote statistically significant (p<0.05) differences between the sites. Note that Orchard was not measured in 2020.



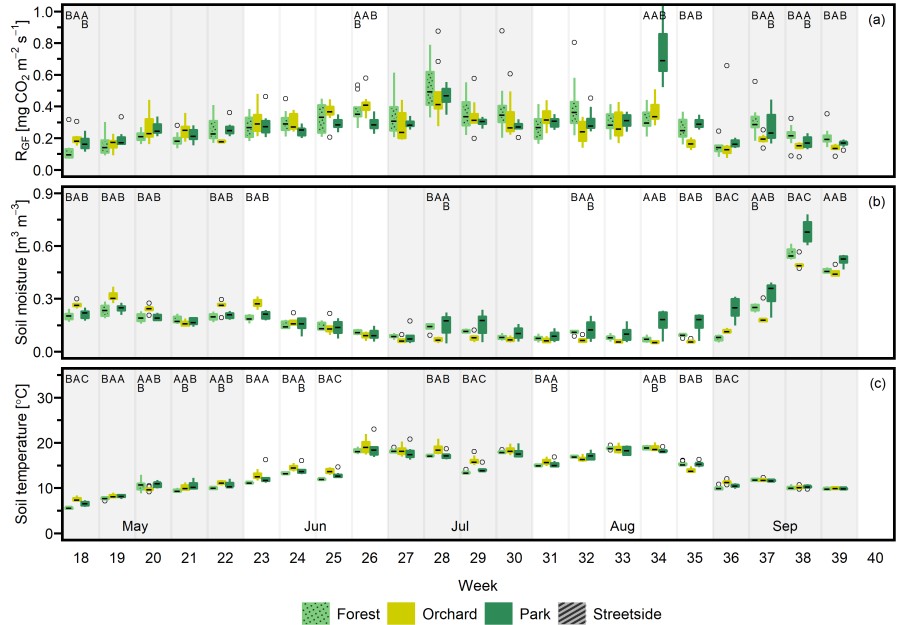

**Figure D2.** a) Soil respiration ($R_{GF}$), b) soil moisture (at 10 cm depth), and c) soil temperature (at 10 cm depth) were measured weekly at three measurement sites (Forest, Orchard, and Park) in 2022. Here, boxes are arranged chronologically by week number, and the sites are always presented in the order that is shown in the legend. Background shading indicates the month. Empty circles are outliers. Letters A-C denote statistically significant ($p<0.05$) differences between the sites. Note that Streetside was not measured in 2022.



*Author contributions.* EK: Conceptualisation, Data curation, Formal analysis, Investigation, Methodology, Visualisation, Writing - original draft, Writing - review & editing. LB: Conceptualisation, Investigation, Methodology, Writing - original draft, Writing - review & editing. LJ: Conceptualisation, Funding acquisition, Project administration, Resources, Writing – review & editing. LK: Conceptualisation, Funding acquisition, Resources, Supervision, Writing - original draft, Writing – review & editing.

*Competing interests.* The authors declare that they have no conflict of interest.

*Acknowledgements.* We would like to thank Yasmin Frühauf, Olivia Kuuri-Riutta, Pinja Rauhamäki, and Jesse Soininen for their help with the manual measurement field work in 2020-2022. We would also like to thank Jarkko Mäntylä and Erkki Siivola from University of Helsinki as well as Juuso Rainne and Timo Mäkelä from FMI for assisting in all technical problems and maintaining our measurement equipment during the campaign. Juha-Pekka Tuovinen, Mika Aurela, Mika Korkiakoski and Helena Rautakoski are acknowledged for their help with the chamber measurement data analysis, Anu Riikonen for her valuable insight into previous research on the topic of urban soil carbon, and Quentin Bell for helping to finalise the language. Finally, we wish to express gratitude to City of Helsinki, Finnish Museum of Natural History (LUOMUS), and Gardening Association for Children and Youth for fruitful co-operation and for allowing us to establish our measurement sites on their property.

This study was supported by the Research Council of Finland (CarboCity, grants no. 325549 and 21527), by the Strategic Research Council working under the Research Council of Finland (CO-CARBON, grants no. 335204 and 335201), in the ACCC Flagship program of the Research Council of Finland (grants no. 337552 and 337549), and the European Union's Horizon 2020 Research and Innovation program (PAUL, grant no. 101037319).





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
