# Peer review of "Soil respiration across a variety of tree-covered urban green spaces in Helsinki, Finland"

_EGUsphere, 2023_

## Author Comment (AC2)

**Authors' reply to referee comments**

We would like to express our gratitude to Referee #2 for their thorough efforts in reviewing our manuscript – thank you! We have considered the comments and will be addressing each of them in the sections below.

With kind regards,

Esko Karvinen and the co-authors

| Referee #2 |
|---|

*Karvinen et al. conducted a field measurement over three consecutive growing seasons to examine soil respiration CO2 fluxes and SOC stocks at four measurement sites in Helsinki. The authors conclude that the observed variation in soil temperature alone was not enough to cause variation in soil respiration rates between the studied green space types, perhaps because the soil moisture conditions were uniform, therefore, irrigation could potentially be a key factor in altering the soil respiration dynamics. The manuscript is well-prepared and easy to follow, and the topic is interesting and important.*

*The mechanisms leading to the varying soil respiration are very complex, for example, in addition to soil temperature and moisture, soil pH and P, K, SOC, and SON are also important, and these soil factors change across different seasons. Rather than total SOC, DOC (dissolved organic carbon) could be more closely related to soil respiration. Besides, the microbial community is key to soil respiration, however, it is not considered in this study. So, I recommend more analysis related to the variation in soil respiration.*

**REPLY:**     We thank the referee for their comment and insight into the topic. We strongly agree on the complexity of soil respiration as a phenomenon and do openly acknowledge that we have not considered all possibly influencing factors in our analyses both due to practical limitations in our field designs and as a result of intentionally focusing this study on the effects of soil temperature, moisture, and SOC. To highlight this in the revised manuscript, we have included remarks of discussion on potential underlying factors, that were not in the focus of this study but could have influenced the results, throughout the text whenever necessary. We appreciate the suggestion to study DOC and the role of the microbial community but, unfortunately, lack such data from our measurement sites. Nevertheless, we have included those aspects among the future research directions outlined in the Discussion of the revised manuscript.

As per the referee's request, we conducted some additional analysis with the available data by calculating the mean soil respiration ($R_{GF}$) fluxes at

the measurement sites separately for each study year and comparing them to the variables in our soil sample dataset (i.e. SOC and SON content and stock, soil density, P, K, pH, and soil particle size classes) to look for any significant correlations. The procedure is documented in the Material and methods and Results sections of the revised manuscript, and the outcome of the additional analysis is discussed in the Discussion.

---

## Author Response (AR1)

**Authors' reply to referee comments, after conducting the revisions**

We would like to express our gratitude to Referees #1 and #2 for their thorough efforts in reviewing our manuscript – thank you! We have considered their comments and will be addressing each of them in the sections below. Simultaneously, we will introduce the changes made in the revised manuscript in **bold** and show excerpts of the revised manuscript in blue.

Any additional changes outside the scope of the referee comments are introduced in the end of this document.

With kind regards,

Esko Karvinen and the co-authors

**Referee #1**

**General comments**

*The authors present results from a temporally extensive dataset providing insights into soil respiration dynamics in urban green spaces. The objectives were to distinguish differences in soil respiration rates measured in different types of tree-covered urban greenspace as well as assessing the impact of the UHI effect and increased irrigation on soil respiration. Results suggest that despite differences in sites, management, tree cover, SOC and soil temperature there were no distinct differences in soil respiration - possibly due to similar soil moisture contents across the sites. Further, model results suggest that while increased irrigation may result in an up to 50% increase in heterotrophic soil respiration an increase in air temperature related to the UHI effect only has a small impact on urban forest and park sites.*

*Overall, the paper is well written with clear objectives and conclusions supported by the results. The use of an ecosystem model to understand UHI effects and the impact of irrigation on soil respiration seems to be an interesting approach and encourages further studies to simulate the urban carbon cycle for different future scenarios. However, there is a need to add further information about the ecosystem model as outlined below.*

**REPLY:**     We are happy to hear that our work was appreciated and would like to thank the referee for their positive and constructive feedback. Next, we will address each of the specific comments and technical corrections.

**Specific comments**

*Re ecosystem modelling - It may help having a paragraph in the introduction introducing the model, what it does, how it is used and what the model inputs/outputs are and if it has been applied for similar applications as in this paper. It also seems there are a lot of assumptions being made about model inputs and it is not always clear how these vary across the different sites and whether these introduce any uncertainties into the model results.*

**REPLY:**   We thank the referee for highlighting the need to improve the subsection "2.7 Ecosystem modelling". In the revised manuscript, we have restructured the whole subsection and paid especial attention to i) starting with a paragraph giving a slightly broader introductory overview of the model and some examples of its recent usage, and ii) clarifying the way the model inputs are described so that it should now be easier to distinguish which of the variables differ between the sites. **This, hopefully now improved, subsection starts at P9 L235 in the revised manuscript.**

However, we feel that the most suitable place for the model introduction is in this specific subsection and not in the actual Introduction section of the manuscript, as was suggested by the referee. **That being said, we have modified the sentence on P3 L89 in the revised manuscript, in which the modelling approach is mentioned for the first time in the Introduction to give more insight into why it was selected as a method to begin with:**

"Additionally, we used process-based ecosystem model trained with the observations to specifically answer research questions 2 and 3 since controlled field experiments are difficult to perform, especially in the case of mature trees. Tested modelling tools are also needed for potential applications such as estimating C fluxes in urban nature in the future climate."

**We have also included one paragraph of discussion on the possible uncertainties arising from the assumptions regarding the modelling setup in the Discussion section of the revised manuscript at P21 L503:**

"We aimed to carefully accommodate our modelling setup to the environmental conditions at the measurement sites to reduce uncertainty in the modelling results but, naturally, there are some possible sources of error stemming from the process. Most of the site-specific parameter values were altered based on observations of, e.g., particle size distribution, soil moisture and LAI, while soil and root depth were estimated based on literature data and the soil depth in the area. Both approaches introduce potential sources of error as observations can have their respective inaccuracies and drawing solely from literature lacks local verification. As an example, errors in the estimated soil and root depths could potentially change the drought response of the sites. Secondly,

since the vegetation phenology at the sites was estimated based on LAI observations, errors in the observations would then in turn be reflected in the accuracy of the modelled phenology. We also assumed that the soil C pools at the simulated sites were in a steady state, which is not necessarily the case in the urban setting. Consequently, such uncertainties should be kept in mind while interpreting the modelling results."

*P5 L126 Measurements were undertaken between 8 AM and 4 PM – did you notice any diurnal variations depending on the time of day the measurements were taken? And if so, how did this affect results?*

**REPLY:** We thank the referee for the relevant question. We think that there could possibly be some diurnal variation in the soil respiration ($R_{GF}$) driven by mainly temperature-dependent processes that are in turn influenced by the daily air (and soil) temperature fluctuation. As the timespan during which our measurements were conducted represents only approximately one third of the whole 24-hour cycle and, therefore, does not contain the extreme points of possible diurnal variation, we did not consider it necessary to account for the time of day itself when conducting our analyses and interpreting the results. Furthermore, disentangling the presence of a diurnal cycle in the measured soil respiration ($R_{GF}$) from the possible simultaneous effects of stochastic phenomena such as precipitation events would be challenging.

To dig a bit deeper into the dataset, we calculated Pearson correlation between the measurement time and the measurement result both for soil respiration ($R_{GF}$) and soil temperature separately for all of the measurement sites (Figure R1). We found statistically significant ($p<0.05$) correlation at Orchard and Park but even there the coefficient of determination ($R^2$) was notably low indicating a subtle trend but a very limited explanatory power of the time of day on the studied variables. There was no significant correlation at the other two sites. Thus, we feel confident in the way we have approached our data and the analyses in this regard.

**As a sidenote, some measurements were actually still conducted between 4 PM and 5 PM, as can be seen from the figure; we corrected the incorrect time at P5 L128 in the revised manuscript.**

[Figure]

**Figure R1.** Manual measurements of soil respiration ($R_{GF}$) and soil temperature were conducted between 8 AM and 5 PM at four measurement sites (Forest, Orchard, Park, and Streetside). Here, all single measurements from 2020-2022 are shown in relation to the time of day they were conducted at. Pearson correlation coefficient of determination ($R^2$) and its respective p-value are reported separately for each measurement site.

*P5 L129 how were measurement points selected within each site?*

**REPLY:**     **We thank the referee for pointing out the need for additional details and have included them in the revised version of the manuscript at P5 L131 onwards:**

"Eight chamber measurement points were systematically selected at each measurement site and the measurements were always performed at these fixed points. Overall, the aim of the selection was to capture the spatial variation within each measurement site by ensuring enough distance between the single measurement points and having some of them located closer to trees than others, some closer to the edge of the green space than others, and so on. The measurement points were established along two parallel transects at Orchard and along two almost parallel transects at Forest and Park. At Orchard, the transects were situated 6 meters apart from one another and the measurement points on each transect were 6 meters apart from each other. At Forest and Park, the transects were situated, on average, 4 meters apart from one another and the measurement points on each transect were, on average, 3 meters apart from each other. As the Streetside site was a less than 2.5 meters wide strip of lawn between a roadway and a sidewalk, there was not enough space for multiple parallel transects. Therefore, the measurement points were situated along a single stretch of 17 meters in such a way that there i) were 1-2 meters in between the single measurement points and ii) they covered the whole width of the lawn strip with some points being closer to the roadway and some to the pavement."

**Additionally, we have added some more information in the "2.3 Ancillary measurements" subsection to clarify the selection of soil moisture measurement points starting at P6 L157:**

"…Six fibreglass access tubes (ATS1, Delta-T Devices) were installed at each site. They were not co-located with the chamber measurement points but were scattered around the measurement site with the aim of capturing the spatial variation within the site. Three readings were obtained…"

*2.7 Ecosystem modelling*

*As mentioned above, this section is not very clear. It may help summarising the general inputs (met data, vegetation data, soil characteristics) and outputs of the model and how these were determined for the forest and park site.*

**REPLY:**     We have completely restructured this particular subsection in the revised manuscript as is discussed in the reply to the very first comment above.

*P9 L249: How was root depth determined? Is this a guess?*

**REPLY:**    We did not have measurement-based data on the tree root depth at the measurement sites. Therefore, root depth was determined utilizing Crow (2005) as a starting point from which the depths were then further adjusted i) based on the estimated total soil layer thickness at our measurement sites, and ii) by comparing with the manual soil moisture measurements. **We thank the referee for pointing out the need to include more details and have added this information to the revised manuscript at P9 L255.**

*P9 L256: the loss rates were modified using the temperature and precipitation data from the FMI Kumpala weather station? How was the size of the litter elements determined?*

**REPLY:**    We thank the referee for pointing out the lack of particularity in this sentence. The model uses 30-day averages of air temperature and precipitation from the model driver dataset to calculate pseudo soil temperature and soil moisture, which are used to dynamically modify the predetermined loss rates of the litter pools, as part of the standard functioning of the model. So, yes, the loss rates were modified with data originating from the FMI Kumpula weather station. We have clarified this in the revised manuscript. **Regarding the litter element sizes, we have added the following to the revised manuscript at P10 L277:**

"The decomposition rate of woody litter is slower than that of green litter. It is assumed that the woody litter elements are larger and therefore decompose at a slower rate. The woody litter has a nominal size of four cm. The rates are reduced by multiplying with a size-dependent factor, which can be defined separately for each woody plant functional type. However, currently the same factor (0.53) is applied to all woody litter. In the case of green litter, the factor is equal to one."

As we have thoroughly restructured this subsection in the revised manuscript, this sentence should now be easier to piece together in that sense as well.

*3.4 Modelled RH dynamics – this seems more like a model validation? It would help having a sentence to introduce the purpose of this section (ie comparing temporal variations of modelled reference to observations?)  (ie Figure 5 excluding modelled irrigation)*

**REPLY:**    We thank the referee for a good suggestion. **In the revised manuscript, we changed the subsection 3.4 title at P15 L364 to "Model performance validation" and added the following sentence in the beginning of it at P15 L365:**

"To validate the model performance, we compared the temporal dynamics of the modelled $R_H$ to the measured $R_{GF}$."

*3.5 This section then describes the results from the UHI and irrigation simulations (Figure 6).*

**REPLY:** Yes, we agree, and have changed the title of the previous section 3.4 in the revised manuscript to help the reader to better grasp the gist of the separate sections, as also discussed in the previous comment.

*P16 L340: Move sentence about impact of the UHI to P18 L411.*

**REPLY:** **We thank the referee for the suggestion but since there already is a sentence with basically identical information content at P18 L411 (line numbers refer the original draft), we opted to simply remove the sentence from P16 L381 in the revised manuscript rather than moving it.**

*P16 L361: SOC stocks were similar to those measured other top layers (0 – 30 cm) but on the lower end compared to studies that measured SOC stocks to 100 cm.*

**REPLY:** We agree and thank the referee for a good remark. **We have re-formulated the beginning of the paragraph at P17 L401 in the revised manuscript accordingly:**

"In comparison to previous research, our measurements of SOC stocks in urban green space (on average 7.37-10.92 kg m$^{-2}$) are similar to those measured in the top layers but on the lower end when compared to studies that measured SOC stocks down to 100 cm depth (Table 4). Differences in sampling depth make straightforward comparison difficult."

*P18 L416: I guess this is not as surprising given the above statement (L398) and previous studies indicating that soil moisture is the main factor controlling urban Rs?*

**REPLY:** That is true – **we have removed the unnecessary word "surprising" from the sentence at P20 L458 in the revised version of the manuscript.**

*Table 4 – Consider adding your results too for direct comparison.*

**REPLY:**    We thank the referee for the remark and **have included our results in the table at P18 in the revised manuscript as per suggestion and modified its caption and footnotes accordingly.**

**Technical corrections**

*Figure 5/6 – the manual measurement points and facets are hard to see in Figure 5 and 6 please use a darker shade or colours to distinguish these.*

**REPLY:**    We were not entirely certain on what the referee was referring to with the facets being hard to see. **Nevertheless, we have adjusted the colors of the manual measurement points and the facet labels (e.g. "2021" and "Park") in the revised manuscript to enhance overall readability of the figures, as can be seen below.** The former versions of the figures are not included in the marked-up manuscript document.

We are happy to make further modifications if what we did now was not of help with what the referee was concerned with in the first place.

[Figure]

**Figure 5.** JSBACH modelled daily heterotrophic soil respiration ($R_H$, left axis) (both reference and irrigation simulation) showed similar temporal dynamics in comparison with the manually measured soil respiration ($R_{GF}$, right axis). Manual measurements are portrayed as mean ± standard deviation, and background shading indicates the study period May–Sep.

[Figure]

**Figure 6.** Daily heterotrophic soil respiration ($R_H$) at Forest and Park was modelled with JSBACH to study the effect of the urban heat island (UHI) and irrigation. During the study period of May-Sep (indicated with background shading), air temperature was increased by 0.5, 1.0, 1.5, and 2.0 °C, and an irrigation algorithm was used to simulate lawn irrigation during dry periods. A reference simulation was conducted separately for both measurement sites (Forest and Park) with the observed local weather conditions of each year.

**Referee #2**

*Karvinen et al. conducted a field measurement over three consecutive growing seasons to examine soil respiration CO2 fluxes and SOC stocks at four measurement sites in Helsinki. The authors conclude that the observed variation in soil temperature alone was not enough to cause variation in soil respiration rates between the studied green space types, perhaps because the soil moisture conditions were uniform, therefore, irrigation could potentially be a key factor in altering the soil respiration dynamics. The manuscript is well-prepared and easy to follow, and the topic is interesting and important.*

*The mechanisms leading to the varying soil respiration are very complex, for example, in addition to soil temperature and moisture, soil pH and P, K, SOC, and SON are also important, and these soil factors change across different seasons. Rather than total SOC, DOC (dissolved organic carbon) could be more closely related to soil respiration. Besides, the microbial community is key to soil respiration, however, it is not considered in this study. So, I recommend more analysis related to the variation in soil respiration.*

**REPLY:**     We thank the referee for their comment and insight into the topic. We strongly agree on the complexity of soil respiration as a phenomenon and do openly acknowledge that we have not considered all possibly influencing factors in our analyses both due to practical limitations in our field designs and as a result of intentionally focusing this study on the effects of soil temperature, moisture, and SOC. **To highlight this in the revised manuscript, we have included remarks of discussion on potential underlying factors, that were not in the focus of this study but could have influenced the results, at the following places:**

**P19 L441 (a new sentence in Discussion)**

"Furthermore, there are also other controlling factors for RS than the ones considered in this study: differences in the soil microbial community (Liu et al., 2018) or the level of dissolved organic carbon (DOC) (van Hees et al., 2005) between the measurement sites, for example, can also have influenced the results."

**P22 L521 (a new paragraph in Discussion built around one previously existing sentence)**

"It is hard to explicitly account for the singular effects of multiple important and co-affecting environmental variables in a field measurement setup like ours, in which the main idea is to measure the studied phenomena of interest ($R_S$) in the naturally varying environmental conditions rather than conducting measurements in a strictly controlled study setup. For example, our soil sampling design did not allow us to effectively delve into the specific effects individual soil characteristics (e.g. levels of various nutrients or the particle size distribution) could have had on $R_S$; collecting separate soil samples from each chamber measurement point would possibly have been a more effective approach for that. However, as the focus of this study was to examine the effects of soil temperature, soil moisture, and SOC, a different sampling design was eventually selected. Investigating the roles of both the soil per se and the soil microbial community on urban $R_S$ would likely prove to be an interesting direction for future research."

**P22 L542 (finetuning the Conclusions)**

"Overall, our findings challenged some of our initial hypotheses, and would encourage further studies on the topic, for example, utilising a measurement site setup with a broader spatial span, more site type replicates, **and a more intricate take on soil characteristics including the soil microbial community.** Based on our results, different soil temperature conditions are likely not the sole explanation for the previously discussed differences in the magnitude of soil respiration between urban and non-urban ecosystems – we cautiously emphasise the role of irrigation and soil moisture and hope to motivate further studies on the topic. We would also tend to agree with Decina et al. (2016) on the

roles of possible organic amendments and the soil itself, especially soil organic carbon, in generating the differences in soil respiration between urban and non-urban ecosystems. **Similarly, soil characteristics are likely an important factor in establishing variation in soil respiration within a city, but disentangling their specific effects from those of soil temperature and moisture was not in the scope of this study.**"

We appreciate the suggestion to study DOC and the role of the microbial community but, unfortunately, lack such data from our measurement sites. **Nevertheless, we have included those aspects among the future research directions outlined in the Discussion of the revised manuscript at P22 L527.**

As per the referee's request, we conducted some additional analysis with the available data by calculating the mean soil respiration ($R_{GF}$) fluxes at the measurement sites separately for each study year and comparing them to the variables in our soil sample dataset (i.e. SOC and SON content and stock, soil density, P, K, pH, and soil particle size classes) to look for any significant correlations. **The procedure and the outcomes are documented in the revised manuscript as follows:**

**P8 L229 (in Material and methods, a new sentence was added while simultaneously forming a new paragraph by re-locating two previously existing sentences)**

"Additionally, we calculated the mean $R_{GF}$ rate at each measurement site separately for each of the study years 2020–2022 and compared them to the site-specific soil characteristics (i.e. SOC and SON content and stock, soil density, P content, K content, pH, and soil particle size classes) by calculating Pearson correlation coefficients."

**P13 L340 (in Results)**

"Additionally, we did not find any statistically significant correlations ($p < 0.05$) when comparing the site-specific yearly mean $R_{GF}$ rates to the respective soil characteristics."

**P22 L523 (in Discussion)**

"For example, our soil sampling design did not allow us to effectively delve into the specific effects individual soil characteristics (e.g. levels of various nutrients or the particle size distribution) could have had on $R_S$; collecting separate soil samples from each chamber measurement point would possibly have been a more effective approach for that. However, as the focus of this study was to examine the effects of soil temperature, soil moisture, and SOC, a different sampling design was eventually selected."

**Topic editor**

*The proposed manuscript presents findings from a temporally extensive dataset, providing insights into soil respiration dynamics and soil carbon stocks in urban green spaces. The aim of the manuscript is timely and the methodology used is adequate to the type of research.*

*The reviewers appreciate the study, and suggested some revisions to improve the description of methods and overall discussion. The authors replied to the reviewers' comments with professionalism, extensively addressing individual comments.*

*The authors should therefore submit a new improved version of the manuscript, as suggested by the referees reports.*

*With the next revision, also please re-name supplement tables according to our standards: https://www.soil-journal.net/submission.html#assets > Supplements.*

**REPLY:**     We thank the topic editor for their kind feedback. We are, however, unsure what is incorrect with the supplement table names. According to the instructions, they should be named S1, S2 and so on, which is what is done in our supplement as well. They were, indeed, incorrect in the very beginning of the submission process, but were already at that point corrected as requested before the preprint was published. Nevertheless, we are happy to make any further modifications if such are needed.

**Other modifications**

**P24 Table A1**

Some of the approximate ages of the measurement sites were incorrect, and they have now been corrected in the revised version of the manuscript as follows:

Park        34 (used to be incorrectly 26)

Streetside   53 (used to be incorrectly 34)

Also, an unnecessary "~" symbol was removed from the age of Orchard, as it is already specified in the caption that the ages are approximations.